Reduced object related negativity response indicates impaired auditory scene analysis in adults with autistic spectrum disorder

Lodhia Veema 1
Brock Jon 2 3
Johnson Blake W. 2 3
Hautus Michael J. 1 m.hautus@auckland.ac.nz
1 Research Centre for Cognitive Neuroscience, School of Psychology, The University of Auckland , New Zealand
2 ARC Centre of Excellence in Cognition and its Disorders , Australia
3 Department of Cognitive Science, Macquarie University , Sydney , Australia
Rojas Don
Electronic publication date: 2014 Feb 25
Publication date: 2014
Volume: 2
Electronic Location ID: e261
Received 2013 Oct 15; Accepted 2014 Jan 15
Copyright: © 2014 Lodhia et al.
Copyright year: 2014
Copyright holder: Lodhia et al.
License: This is an open access article distributed under the terms of the Creative Commons Attribution License, which permits unrestricted use, distribution, reproduction and adaptation in any medium and for any purpose provided that it is properly attributed. For attribution, the original author(s), title, publication source (PeerJ) and either DOI or URL of the article must be cited.
License URL: https://creativecommons.org/licenses/by/4.0/

Keywords: Autism, Auditory scene analysis, Object related negativity, Event related potential, Binaural processing, Electroencephalography, ORN, P400, Dichotic pitch

Funding: Australian Research Council (ARC) Centre of Excellence for Cognition and its Disorders CE110001021 ARC Australian Research Fellowship DP098466 The work of BWJ and JB was supported by the Australian Research Council (ARC) Centre of Excellence for Cognition and its Disorders (CE110001021): http://www.ccd.edu.au. JB was supported by an ARC Australian Research Fellowship (DP098466). The funders had no role in study design, data collection and analysis, decision to publish, or preparation of the manuscript.

==============================
Auditory Scene Analysis provides a useful framework for understanding atypical auditory perception in autism. Specifically, a failure to segregate the incoming acoustic energy into distinct auditory objects might explain the aversive reaction autistic individuals have to certain auditory stimuli or environments. Previous research with non-autistic participants has demonstrated the presence of an Object Related Negativity (ORN) in the auditory event related potential that indexes pre-attentive processes associated with auditory scene analysis. Also evident is a later P400 component that is attention dependent and thought to be related to decision-making about auditory objects. We sought to determine whether there are differences between individuals with and without autism in the levels of processing indexed by these components. Electroencephalography (EEG) was used to measure brain responses from a group of 16 autistic adults, and 16 age- and verbal-IQ-matched typically-developing adults. Auditory responses were elicited using lateralized dichotic pitch stimuli in which inter-aural timing differences create the illusory perception of a pitch that is spatially separated from a carrier noise stimulus. As in previous studies, control participants produced an ORN in response to the pitch stimuli. However, this component was significantly reduced in the participants with autism. In contrast, processing differences were not observed between the groups at the attention-dependent level (P400). These findings suggest that autistic individuals have difficulty segregating auditory stimuli into distinct auditory objects, and that this difficulty arises at an early pre-attentive level of processing.

Introduction

Autism is a developmental disorder that is defined and diagnosed in terms of impairments in social interaction and communication co-occurring with restricted behaviours and interests (American Psychiatric Association, 1994; American Psychiatric Association, 2013). In addition to these core diagnostic ‘symptoms’, many individuals with autism also experience hyper- or hypo-sensitivities in visual, auditory, and tactile domains (Talay-Ongan & Wood, 2000; Grandin & Scariano, 1986; Rosenhall et al., 1999). Indeed, the recent revision of the DSM-5 (2013) diagnostic criteria for Autism Spectrum Disorder now makes explicit reference to sensory symptoms. Atypical auditory processing is particularly well documented. Many autistic individuals experience a distressing hyper-reactivity to noise (Grandin & Scariano, 1986; Rosenhall et al., 1999) and several studies have reported that autistic individuals have difficulty extracting relevant auditory information (i.e., speech) in the presence of competing background noise (Boatman et al., 2001; Alcantara et al., 2004; Teder-Salejarvi et al., 2005; Groen et al., 2009).

In the current study, we investigated auditory processing in autism within the context of Bregman’s (1990) auditory scene analysis framework. According to Bregman, auditory perception involves grouping the incoming acoustic information into distinct auditory “objects” that correspond to inferred events in the listener’s environment. This grouping occurs across time, space, and frequency and is determined by gestalt principles (such as similarity and temporo-spatial proximity) as well as attention and top-down effects of prior knowledge. Traditionally, auditory scene analysis has been investigated using behavioural methods in which participants report what they perceive as a function of stimulus manipulations. However, such methods are likely to be inappropriate for individuals with developmental disorders such as autism, who may be unable to provide an accurate introspective report of their perceptual experience. For this reason, investigations of auditory scene analysis in autism have measured auditory grouping indirectly via the measurement of brain responses.

In a 2005 study, Teder-Salejarvi et al. reported that, amongst individuals with autism, brain responses to sounds emanating from attended versus ignored spatial locations were indistinguishable. The authors concluded that the ability to focus auditory attention in complex acoustic environments is impaired in autism. However, this result could also indicate a problem with low-level perceptual segregation of the two sources. Subsequently, Lepistö et al. (2009) investigated auditory streaming using the mismatch negativity (MMN) paradigm. Adults with autism evidenced a typical MMN response to pitch deviants in a sequence of tones. However, this effect was eliminated when a separate stream of much higher tones was overlain, suggesting that the participants with autism did not segregate the sounds into separate auditory streams.

The current study investigated concurrent auditory segregation in adolescents and young adults with autism via the dichotic pitch paradigm. Dichotic pitch refers to the perception of pitches from stimuli that do not contain monaural cues to pitch (Bilsen, 1976; Cramer & Huggins, 1958; Dougherty et al., 1998). Time-shifted dichotic pitch is created by presenting to each ear copies of broadband noises that have identical spectra but contain interaural time delays across a narrow frequency band. The frequency band containing the delay becomes perceptually separated from the remaining noise and is heard as a pitch with a tonal quality that is related to the centre frequency of the narrow frequency band (Johnson, Hautus & Clapp, 2003; Hautus & Johnson, 2005). Because the time shift has no effect on the spectral content of the stimuli, any differential response can be assumed to reflect the cortical processes underlying auditory segregation (Hautus & Johnson, 2005; Johnson et al., 2004; Hautus, Johnson & Colling, 2009).

Our previous research using such stimuli has demonstrated that perception of dichotic pitch is associated with a negative ERP component with a latency of about 150–250 ms (Hautus & Johnson, 2005; Clapp, Johnson & Hautus, 2007; Johnson et al., 2007). This Object Related Negativity (ORN) was originally described by Alain, Arnott & Picton (2001) in the context of mistuned harmonics. It arises with or without attention to the auditory stimuli and is therefore assumed to represent a neurological marker of the pre-attentive stage of auditory scene segregation (Alain, Arnott & Picton, 2001; Johnson & Hautus, 2010).

A magnetic counterpart, the mORN, has also been found using magnetoencephalography (MEG) (Johnson & Hautus, 2010; Johnson et al., 2004; Alain & McDonald, 2007). In a recent MEG study, we found that children with autism failed to show an mORN to dichotic pitch stimuli, suggesting a failure of auditory segregation (Brock et al., 2013). However, results were inconclusive as the magnitude of the ORN was not significantly smaller than that evidenced by age-matched typically developing children.

The current study built on our earlier MEG study, using EEG to investigate the ORN. Rather than testing children, we tested young adults with autism, thereby allowing us to administer many more trials and achieve more reliable responses. Moreover, brain responses of adults are likely to be more consistent across individuals. Auditory evoked responses are typically mature by late adolescence (Mahajan & McArthur, 2012) and studies comparing adolescents, young adults and middle-aged adults have found no evidence of developmental change in the ORN (Alain, Arnott & Picton, 2001; Alain et al., 2003; Alain & McDonald, 2007). To maximize the ORN response, we used “lateralized” dichotic pitch stimuli, whereby the broadband noise is also time-shifted in a direction opposite the narrow frequency band. In this case the segregation of pitch and noise is enhanced such that the listeners perceive the broadband noise lateralized to one side of auditory space and the pitch lateralized to the other side (see Fig. 1; Johnson & Hautus, 2010). This contrasts with the stimuli in our MEG study in which the pitch was lateralized but the residual noise was presented without an interaural timing difference and was therefore perceived as emanating from the centre of space.

Figure 1 Schematic representations of the dichotic pitch stimuli.

These representations indicate the nature of the percept associated with the four stimulus configurations. Panels A and B show the No Pitch (or control) stimuli that lead to the perception of a noise lateralized to one side of auditory space. Panels C and D show the Pitch (or target) stimuli which also lead to the perception of a noise, but in addition, a pitch is perceived lateralized to the side opposite the noise. (Noise represented by ### and Pitch represented by ♪).

As a final point of difference, we added a behavioural task in which participants were required to indicate via button press whether or not they heard the pitch sound. This contrasts with Brock et al. (2013) in which the participants were instructed to ignore the stimuli whilst watching a sound-attenuated movie. This allowed us to directly compare behavioural and electrophysiological indices of auditory perception. Previous studies have indicated that the addition of a behavioural task elicits a positive component, termed the P400, with a latency of about 400–500 ms and, like the ORN, the P400 can be produced by ITD and inharmonicity. Unlike the ORN, the P400 is attention dependent, occurring only when participants are actively listening to (and discriminating between) stimuli. It is therefore thought to reflect the decision-making process related to the parsing of the incoming sound into concurrent perceptual objects (Alain, Arnott & Picton, 2001; Hautus & Johnson, 2005).

Methods

Participants

Participants were 16 individuals with an Autism Spectrum Disorder (ASD) and 16 typically-developing (TD) individuals. A further 5 participants with ASD were excluded because they were unable to satisfactorily discriminate dichotic pitch during the practice phase (see details below). The two groups of 16 were matched on gender, age (±2 years), and handedness, determined by the Edinburgh Handedness Inventory (Oldfield, 1971).

Participants in the ASD group were recruited via adverts posted at Autism NZ, Altogether Autism, Autism House, Centre for Brain Research, and The University of Auckland. Participants gave their informed written consent, and all procedures were approved by The University of Auckland Human Participants Ethics Committee (Ref: 2009/537).

Exclusion criteria for the ASD participants included a co-morbid Axis 1 disorder and relevant Axis 3 diagnosis, hearing deficits and pharmacological treatment. For participants in the TD group, the exclusion criteria included personal or family history of neurological or psychiatric disorders, hearing deficits, and pharmacological treatment. Further inclusion criteria for both TD and ASD groups were (1) normal auditory acuity — hearing thresholds ≤25 dB HL, as assessed by an audiogram (Amplitude T-Series, Otovation, LLC, USA) for the standard range of 250–8000 Hz; (2) a full-scale mental ability score whose lower confidence bound was ≥80; and (3) passing a pre-screening assessment demonstrating an ability to detect dichotic pitch.

All participants in the ASD group had been given a clinical diagnosis of autistic disorder (N = 3) or Asperger’s disorder (N = 13) according to DSM-IV. Diagnoses were made by a clinical psychologist or paediatrician. As a further check, we determined that all participants met the cut-off for ASD on the Social Communication Questionnaire (SCQ – Lifetime scale ≥15), which was completed by a parent or guardian at the first study meeting. The SCQ is based on the Autism Diagnostic Interview-Revised, with which it has good agreement (Bishop & Norbury, 2002).1

Table 1 summarizes the demographic and behavioural test results for both groups. No group differences were found for verbal or combined IQ as measured using the Wechsler Abbreviated Scales of Intelligence (Wechsler, 1999). A group difference was found for performance IQ, nevertheless the ASD group performed above average for their age group.

Table 1 Demographic and cognitive characteristics of the TD and ASD groups.

Measure	ASD (N = 16)	TD (N = 16)	Range	Independent t-test	
	M (SD)	M (SD)	Min	Max	t	df	p	
Age (years)	22.19 (5.99)	22.69 (5.20)	16	34	0.59	30	.80	
Handedness 100% = right	75.69 (54.70)	68.62 (62.77)	−100	100	−0.34	30	.74	
Verbal IQ	119.50 (18.69)	118.38 (14.89)	84	140	−0.19	30	.85	
Performance IQ	107.25 (13.76)	116.25 (9.95)	72	131	2.12	30	.04	
Combined IQ	114.75 (16.64)	120.31 (12.27)	79	137	1.08	30	.29	
SCQ	23.06 (5.22)	–	15	33	–	–	–	

Stimuli

Two independent broadband Gaussian noise bursts, each 500 ms in duration, were constructed at a sampling rate of 44.1 kHz, using LabVIEW software (National Instruments, Austin, Texas, USA). One noise burst was bandpass filtered with a centre frequency of 600 Hz and a bandwidth of 20 Hz using a fourth-order Butterworth filter. The other noise burst was notch filtered using the same filter characteristics. A copy was made of both noises (bandpass and notch), one copy of each type for each ear. For the target stimulus (noise plus pitch; two auditory objects) opposing temporal delays (±500 µs) were applied to the bandpass- and notch-filtered noises so that the resulting combination would create a noise lateralized to one side of auditory space and a pitch to the other side of auditory space. For control stimuli (noise alone; one auditory object) both the bandpass- and the notch-filtered noise were temporally delayed (500 µs) to the same ear (Fig. 1), resulting in noise lateralized to one side of space. The notch- and bandpass-filtered noise processes within each auditory channel were recombined, producing two spectrally flat noise processes, which were again bandpass filtered (fourth-order Butterworth filter) with a centre frequency of 600 Hz and bandwidth of 400 Hz. The stimuli were windowed with a cos2 function with 4 ms rise and fall times. The auditory stimuli were generated on two-channels of a 16-bit converter (Model DAQPad 6052E; National Instruments, Austin, TX). Programmable attenuators (Model PA4; Tucker-Davis Technologies, Alachua, FL) set the binaural stimuli to yield 70 dB SPL from insert earphones at the ear. (ER2; Etymotic Research Inc., Elk Grove Village, Illinois, USA).

Behavioural task

On each trial, participants indicated on a button box whether the stimulus presented consisted of one or two auditory objects. In an initial practice session, prior to EEG recording, participants completed four 100-trial blocks with feedback received after each trial. Five of the original 21 participants with ASD did not reach the criterion of 69 percent correct (approximately d′ = 1; cf. Macmillan & Creelman, 2005, p. 9) in the practice session and were therefore excluded from the EEG part of the study because they could not sufficiently discriminate between the two types of stimuli.

During the EEG recording, the task was similar, except that no feedback was given and the trial timed out after 1500 ms if no response was made. The inter-stimulus intervals were drawn from a rectangular distribution between 2000 ms and 3400 ms. Participants completed four blocks of 256 trials, each of which took approximately 13 min to complete. Short breaks were given after each block.

Electroencephalography

EEG recordings were conducted in an electrically shielded room (Model L3000; Belling Lee, Enfield, England) using 128-chanel Ag/AgCl electrode nets (Tucker, 1993; Electrical Geodesics Inc., Eugene, Oregon, USA). EEG was recorded continuously (250-Hz sample rate; 0.1–100 Hz analogue bandpass) with Electrical Geodesics Inc. amplifiers (200-MΩ input impedance). Electrode impedances were kept below 40 kΩ, an acceptable level for this system (Tucker, 1993). Common vertex (Cz) was used as a reference. During the EEG, participants were asked to fixate on a cross, presented on a computer screen.

Data analysis

EEG files were segmented into 750 ms epochs (including a 100 ms pre-stimulus baseline) during which all ocular artifacts were corrected (Jervis et al., 1985). Trials with channels marked as bad were dropped from the averaging process. 98% of trials remained for analysis from each group. Given that the ORN is elicited regardless of whether a task is performed, all trials were included, irrespective of response accuracy. ERPs were re-referenced to the average reference. ERPs from individual participants were combined to produce grand-averaged ERPs for each condition. Grand averaged data were then digitally filtered with a zero-phase-shift 3-pole Butterworth filter (0.1–30 Hz; Alarcon, Guy & Binnie, 2000) and then re-referenced to the mean.

For statistical analysis, the electrode clusters of interest for the ORN and the P400 components were selected by combining all 32 participants’ data for the No Pitch and the Pitch conditions. These grand averaged waveform topographic maps were then used to select a symmetrical cluster of electrodes that showed the greatest difference in mean amplitude between the No pitch and the Pitch conditions (left hemisphere electrodes: 7, 12, 13, 20, 28, 29, 30, 31, 37; right hemisphere: 5, 80, 87, 105, 106, 111, 112, 117, 118). For each participant we then averaged across these channels to calculate a Pitch and a No-Pitch waveform. Time windows for the ORN and P400 components were determined based on the full width half max of the difference waveform for the combined group (N = 32). For each participant, the magnitude of the two components was calculated as the area under the curve in the difference waveform.

As Kilner (2013) has recently pointed out, selecting channels and time windows based on the observed peaks inflates the likelihood of false-positives in the within-subjects effect (i.e., it increases the likelihood of finding a main effect of Condition when none exists). However, our aim was not to replicate the numerous previous studies demonstrating the existence of the ORN and P400 but rather to determine whether the components differed in magnitude across groups. Because our choices were all made based on the data averaged across both groups (and because the groups were of equal size), they should not increase the likelihood of a false positive group difference.

Results

Behavioural performance

ANOVA revealed a main effect of Group, (F(1, 30) = 12.75, p < .001, ηp2=.298), indicating that the TD group obtained a higher percentage correct score (87.38%) than the ASD group (70.38%).

Event-related potentials

Figure 2 shows the ERP waveforms for Typically Developing and ASD participants in response to Pitch and No Pitch (Control) stimuli. Typically developing participants showed an increased negativity (ORN) to the Pitch stimuli, coincident with the P2 and N2 peaks. This was followed by an increased positivity P400 at around 400 ms. Waveforms for participants with ASD were similar overall, but there was little evidence of a differential response to Pitch and No Pitch stimuli.

Figure 2 Event related potential waveform graphs.

Grand averaged ERP (−100–750 ms) graphs of the No Pitch and Pitch stimuli for the TD group and ASD group. Shaded regions indicate the time windows used for calculating the ORN (168–284 ms) and P400 (404–520 ms).

For the ORN time window, ANOVA confirmed a more negative response to Pitch compared with the No Pitch stimuli (F(1, 30) = 34.87, p < .001, ηp2=.538). There was no main effect of Group (F(1, 30) = 0.79, p = .382, ηp2=.026). However, as predicted, there was a significant Pitch × Group interaction (F(1, 30) = 8.66, p = .006, ηp2=.224), with a considerably larger effect of Pitch in the TD group (see Fig. 3). Follow-up t-tests (two-tailed) indicated that the TD group showed a significant ORN (t(15) = −6.43, p < .001) but the ASD group did not (t(15) = −2.04, p = .059).

Figure 3 Association between behavioural performance during the EEG recording and the magnitude of the ORN.

Panel B shows a regression line (and confidence intervals) fitted to the ASD data (circles). Boxplots show the distributions of behavioural performance (Panel C) and ORN (Panel A).

Figure 3 also indicates the presence of an outlier in the TD group, with an ORN (−0.72 µV ) that was considerably larger than that of the other participants. We therefore repeated the analyses with the outlier excluded. Critically, the Pitch x Group interaction remained significant (F(1, 29) = 7.31, p = .011, ηp2=.201) indicating that the group differences were not driven solely by this outlier.

Pearson’s correlation analyses revealed that, within the ASD group, better behavioural performance during the EEG recording was associated with a more negative ORN (r(16) = −.567, p = .022). In other words, individuals with ASD who performed well on the task tended to show a typical ORN, whereas those who performed poorly demonstrated a reduced ORN (Fig. 3). Within the TD group, the correlation was in the same direction but fell well short of significance (r(16) = −.314, p = .237), perhaps reflecting ceiling effects on performance.

Further correlation analyses (see Table 2) showed no association between ORN magnitude and either age, scores on the Social Communication Questionnaire, verbal IQ, or performance IQ within the ASD group (minimum p = .34). Similar analyses of the TD group revealed a significant correlation between the ORN and verbal IQ (r(16) = .691, p = .003) but this became non-significant when the outlier was excluded (r(15) = .500, p = .057). All other correlations were non-significant, with or without the outlier.

Table 2 Correlations for each group between electrophysiological measures (ORN and P400) and participant demographics and accuracy.

Measure	ORN	P400	
	ASD	TD	ASD	TD	
Accuracy	−.567*	−.314	.222	−.187	
Age	.253	.046	−.341	.313	
Verbal IQ	.073	.691**	.496	.295	
Performance IQ	−.049	.226	.393	.335	
SCQ	−.148	N/A	−.148	N/A	
Notes.

* p < .05.

** p < .01.

Results for the P400 component were less clear-cut. ANOVA confirmed a more positive response to Pitch compared with the No Pitch stimuli (F(1, 30) = 5.02, p = .033, ηp2=.143). There was again no main effect of Group (F(1, 30) = 0.01, p = .981, ηp2=.000) but, unlike for the ORN, there was no Pitch × Group interaction (F(1, 30) = 0.21, p = .650, ηp2=.007). Follow-up t-tests (two-tailed) indicated that neither the TD group (t(15) = 1.79, p = .094) nor the ASD group (t(15) = 1.36, p = .195) showed a significant effect of Pitch when considered in isolation. Correlations between the P400 and measures of behavioural performance were not significant in either group, although there was a marginally significant association with verbal IQ (r(16) = .496, p = .051). Given the large number of correlations performed, it would be unwise to draw any conclusions based on this finding.

Discussion

Auditory Scene Analysis provides a useful framework for understanding atypical auditory perception in autism. Specifically, a failure to segregate the confusion of incoming auditory energy into distinct auditory objects might explain the aversive reaction autistic individuals have to certain auditory stimuli or environments. Our prediction in this study was that autistic individuals would evidence a reduced ORN, indicating a failure to segregate the dichotic pitch stimuli into spatially separate auditory objects. This proved to be the case. Where TD participants showed a significant ORN, the effect was reduced in adults with ASD, who did not themselves show a significant ORN.

As in previous studies, we focused on electrophysiological measures of auditory perception in order to avoid potential confounds such as task understanding and attention that might affect performance on behavioural tasks. However, there was, in fact, substantial agreement between behavioural and electrophysiological measures both at the group and the individual level. This indicates that, in the high-functioning adults tested here, the behavioural performance is a good indicator of underlying perceptual processes, and that together the two measures provide converging evidence for atypical perception, at least in a subgroup of individuals with ASD.

These results are also broadly consistent with our previous study in which we failed to find a significant ORN in a group of autistic children (Brock et al., 2013). The current results are, however, more compelling insofar as they revealed a significant group by condition interaction, which was only a trend in the earlier study.2 It is not clear which of the various methodological differences might explain this difference in outcome. The current study used EEG rather than MEG, used lateralized noise rather than centralized noise, included more participants and more trials, and involved adults rather than children. Any or all of these differences could be relevant. Alternatively, given the variation in both the ORN and behavioural performance within our ASD group, as well as the inherent heterogeneity in the wider ASD population, results could simply reflect differences in sampling across the two studies. Also of potential significance is the absence of “gold standard” tests for diagnosing adults with autism, and that diagnoses of autism were made by several qualified professionals.

It is also difficult to be sure at this stage to what extent these findings are specific to the dichotic pitch paradigm or reflect auditory segregation more generally. Our ongoing research looks to address this issue by using other auditory stimuli that also produce an ORN. That being said, participants with ASD were all significantly above chance in the practice session, indicating that they were at least able to detect the inter-aural timing differences that gave rise to the dichotic pitch perception. The reduced ORN in their response suggests that, even though they were able to detect some difference between the pitch and control stimuli, their auditory systems did not fully segregate these two sound qualities (noise and pitch) into separate auditory objects. Rather, they are more likely to perceive a single auditory object that has both noise- and pitch-like qualities. The distinction in the qualities of this single object allows the behavioural task to be completed successfully; albeit with lower performance than the TD participants.

This interpretation would also be consistent with the absence of group differences in the later P400 component, which is thought to index the task-based decision. However, caution is required in interpreting the P400 responses given that neither group evidenced a significant P400 effect on their own, and that the P400 response did not correlate significantly with behavioural performance. Thus, it may simply be the case that the P400 response is unreliable, or that its latency or spatial distribution varies across individuals, meaning that we were unable to extract a measure of the P400 size that actually reflected the strength of the underlying neurophysiological processes.

Our working hypothesis, therefore, is that ASD individuals have (or are more likely to have) difficulties in the segregation of auditory stimuli into distinct auditory objects. This ability is known to begin in infancy (Folland et al., 2012; Dermany, 1982; McAdams & Bertoncini, 1997) and continues to improve in conjunction with growth of neuronal connectivity in adolescence (Smith & Trainor, 2011). Reduced ability to filter out and process multiple sounds may, therefore, be attributed to atypical brain development and growth in ASD. Source modelling suggests that the neural generators of the ORN are located in the posterior supratemporal plane for dichotic pitch stimuli (Hautus & Johnson, 2005), consistent with the view that the planum temporale neural network has a functional role in concurrent sound segregation (Alain, Arnott & Picton, 2001; Griffiths & Warren, 2002). Of note, there have been several reports that individuals with ASD have a smaller planum temporale compared to typically developing individuals (Rojas et al., 2002; Rojas et al., 2005) although, without MRIs for the current participants, this remains speculative.

Further research is therefore required to determine how common the deficits in auditory object processing are within the ASD population, and whether they relate at the individual level to atypical perceptual experiences. In particular, our study specifically concentrated on high functioning adults. It is unclear whether we would find similar pre-attentive processing difficulties with other ASD profiles such as younger children and lower functioning individuals. Some sub-groups within the autistic spectrum may have very different auditory perceptual experiences to those tested here.

It also remains to be established how specific these difficulties are to ASD. In a recent study, we found no difference in the ORNs generated by typically developing children and those with specific reading difficulties (Johnson et al., 2013). There are, however, many other conditions associated with atypical auditory processing, and affected individuals might show effects similar to those with autism (e.g., Elsabbagh, Cohen & Karmiloff-Smith, 2010; Goll, Crutch & Warren, 2010). These caveats notwithstanding, the current study adds to the growing body of evidence that atypical auditory perception associated with autism may be understood in terms of differences in auditory scene analysis.

The authors gratefully acknowledge the assistance and support from Autism NZ, Altogether Autism, Autism House, the participants and their families for supporting this research, and Ms Chaturangi Nelumdeniya for assistance with the collection of data.

Additional Information and Declarations

Competing Interests

Author Contributions

Human Ethics

1 A limitation of the current study (and indeed all other studies of adults with autism) is the lack of an “objective” verification of autism diagnosis. The Autism Diagnostic Observation Schedule (Lord et al., 1999) is considered by some researchers to be the gold standard for autism diagnosis. However, studies suggest that in adults it fails to discriminate between autism and other conditions such as schizophrenia that have overlapping symptoms (Bastiaansen et al., 2011).

2 In our previous MEG study, analyses were conducted by a bootstrapping analysis of the difference waveforms. To allow a more direct comparison with the current study, we re-analysed the MEG data, calculating the mean amplitude of the source waveform for the right hemisphere between 250 and 360 ms. This choice was based on analysis of data from a larger sample of typically developing children (Johnson et al., 2013) which showed a significant ORN in this window. Consistent with the current study, we found no ORN in the ASD group (t(9) = 0.23, p = .827), but the group by condition interaction was also non-significant (F(1, 18) = 1.48, p = .239, ηp2=.076.

JB is an Academic Editor for PeerJ. There are no other competing interests.

Veema Lodhia and Michael J. Hautus conceived and designed the experiments, performed the experiments, analyzed the data, contributed reagents/materials/analysis tools, wrote the paper.

Jon Brock and Blake W. Johnson conceived and designed the experiments, analyzed the data, wrote the paper.

The following information was supplied relating to ethical approvals (i.e., approving body and any reference numbers):

The University of Auckland Human Participants Ethics Committee: Ref: 2009/537.

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
