# Peer review of "Reduced object related negativity response indicates impaired auditory scene analysis in adults with autistic spectrum disorder"

_PeerJ, doi:10.7717/peerj.261_

## Round 0.1 · original submission · Minor Revisions

I agree with the reviewers that the article is fundamentally sound. In your response to the reviews, please pay particular attention to the point that both reviewers made concerning the group difference in behavioral performance and the relationship between ORN and behavior. In addition, please clarify as well the point about the selection of the electrode cluster. Given emerging statistical concerns over picking electrode regions of interest (see Kilner JM Clinical Neurophysiology, 124 (10) , 2062-2063, 2013), please be clear about the criterion used.

Reviewer 1 ·

Basic reporting

No comments

Experimental design

1) The primary concern regards the group differences in behavioral performance. In the introduction, it says that the inclusion of the behavioral task would allow for direct comparisons of the behavioral and electrophysiological indices of auditory perception. However, these comparisons were not undertaken. Given that there were behavioral differences, these differences are a potential confound. Is there a correlation between performance scores and the EEG measures? If so, performance should be included as a covariate in the statistical analyses. Or alternatively, only accurate trials should be included.

2) Could you clarify a few things about the methods?
a) After artifact correction, how many trials remained for each of the groups? Were there any group differences in the amount of trials included?

b) How was the cluster of electrodes selected? It is unclear what “the greatest mean amplitudes in the grand-averaged waveform” means.

c) For each subject, how was the difference waveform computed? Was the data from all the electrodes in the two clusters of electrodes averaged in the pitch and no pitch condition and then these averages used to compute the difference waveform?

Validity of the findings

No comments

Additional comments

No comments

Reviewer 2 ·

Basic reporting

No comments

Experimental design

No comments

Validity of the findings

No comments

Additional comments

This is a well written manuscript investigating auditory perception in ASD. It applies an established experimental paradigm using lateralized dichotic pitch stimuli to elicit distinct ERP components representing pre-attentive and attentive aspects of auditory segregation. The study follows programmatically from a series of investigations using complementary methods and thoughtfully distinct experimental paradigms. The manuscript represents a meaningful contribution to the literature on a topic of relevance to ASD. The suggestions below are offered to correct minor weaknesses and to strengthen the manuscript by capitalizing upon the breadth of data collected.

1. The introduction might be revised to reflect that, under DSM-5, sensory hyper- or hypo-sensitivities are diagnostic symptoms (Lines 20-24).
2. Diagnostic characterization is not consistent with gold standard research approaches, and this should be acknowledged as a weakness of the study. Nevertheless, the detection of group differences suggests the clinical group differed from the typical participants meaningfully.
3. Please clarify whether participants were required to attain any performance criteria during practice trials.
4. Please provide the actual time windows used for ERP extraction. It would be helpful to have the shaded portions in Figure 2 correspond to these windows.
5. On Line 223, it is often preferable to state “individuals with autism” rather than “autistic individuals”.
6. The manner in which this study builds from prior work is a strength. However, the authors raise several changes that could contribute to the current findings: recoding method (EEG vs. MEG), experimental method (lateralized pitch and noise vs. lateralized pitch alone), sample size (10 vs. 16), age (children versus young adults), and number of trials. It would be helpful to consider these factors individually in interpreting the results. Some specific analyses that would enrich the study include:
a. Contrasting effect size in this study versus the MEG study. This would help to clarify whether it is the larger sample size or other factors that account for significance. If the effect size is the same, it suggests sample size is key here. If not, it suggests that age, number of trials, or dual lateralization might be of particular relevance.
b. Examining the influence of age. This is an important departure from prior work and several pieces of additional information would be helpful.
i. Include Min and Max in Table 1 (also include for other characteristics).
ii. The SD suggests considerable age range. Please provide more information about the development of the auditory processes across the age span included in this study (development is addressed briefly on Lines 247-251).
iii. It would also be informative to explore the relationship between ORN and age by covarying in the ANOVA or looking at correlations. This would help to clarify potential explanations for the stronger results here than in prior work.
c. Given the group differences in performance IQ, please examine ORN as a function of this factor to confirm that diagnostic status, rather than PIQ, best accounts for the interaction at the ORN.
d. Given the group difference in behavioral task performance, please report the relationship between ORN and task performance. This would also provide stronger justification for the suggestion on Line 240 that reduced ORN in ASD is related to poorer performance on the behavioral task.
e. Examining the relationship between behavioral performance and P400 might also clarify the absence of expected effects at this component.
f. Please report any relationships (or lack thereof) between autism symptomatology (i.e., SCQ score) and behavioral performance and ERP response.
7. Please address in the discussion the failure to detect predicted effects at the P400.

---

## Round 0.2 · Minor Revisions

1. While I agree that there are problems with the two instruments mentioned in particular (ADOS and ADI) for adult participants, the reviewer’s comment is larger than the instrument specifics. The article does not address the qualifications of the professional used for diagnosis, which the rebuttal claims is their gold standard. Please be clear in your revised manuscript whether a single individual did the diagnoses and what his/her qualifications were (e.g., clinical psychologist, pediatrician, etc.). Also, if more than one individual did the diagnosis, please add this as a limitation to the discussion, as reliability of diagnosis would be questionable in this case. One advantage to the ADI and ADOS is the reliability of the spectrum diagnosis, although it is conceded that the specific DSM-IV category is not reliably obtained even with those so-called gold standards. In addition, please state who the informant(s) was for the SCQ. As stated by the authors in the rebuttal, the SCQ is derived from the ADI, which typically asks questions about remote events from an adult’s childhood. So, it stands to reason that it suffers the same limitations as the ADI.

2. Providing effect sizes is a reasonable request that I am inclined to agree with. The nuances of effect size interpretation can certainly be discussed in the article from your perspective as the authors. I agree that with the many differences between the two studies, it is not a simple matter to compare the two for sample size limitations. But it seems better to have the data so that readers can then form their own informed opinion. Please provide estimates from the MEG study and the current study in the revision.

---

## Round 0.3 · accepted · Accept

Thank you for your thoughtful consideration of the reviewers' and editor's comments. I find the revised manuscript acceptable for publication in PeerJ.